# Enhancing image captioning with depth information using a Transformer-based framework

## Abstract

Captioning images is a challenging scene-understanding task that connects computer vision and natural language processing. While image captioning models have been successful in producing excellent descriptions, the field has primarily focused on generating a single sentence for 2D images. This paper investigates whether integrating depth information with RGB images can enhance the captioning task and generate better descriptions. For this purpose, we propose a Transformer-based encoder-decoder framework for generating a multi-sentence description of a 3D scene. The RGB image and its corresponding depth map are provided as inputs to our framework, which combines them to produce a better understanding of the input scene. Depth maps could be ground truth or estimated, which makes our framework widely applicable to any RGB captioning dataset. We explored different fusion approaches to fuse RGB and depth images. The experiments are performed on the NYU-v2 dataset and the Stanford image paragraph captioning dataset. During our work with the NYU-v2 dataset, we found inconsistent labeling that prevents the benefit of using depth information to enhance the captioning task. The results were even worse than using RGB images only. As a result, we propose a cleaned version of the NYU-v2 dataset that is more consistent and informative. Our results on both datasets demonstrate that the proposed framework effectively benefits from depth information, whether it is ground truth or estimated, and generates better captions. Code, pre-trained models, and the cleaned version of the NYU-v2 dataset will be made publically available.

## 1 Introduction

In recent years, there has been a significant research effort in the integration of vision and language to bridge the gap between the computer vision and natural language processing domains. Vision-Language tasks include processing a picture and a text that is related to it simultaneously. Several vision-language tasks have achieved significant progress, including image-text matching (Lee et al., 2018; Yu et al., 2014), visual question answering (Antol et al., 2015; Yu et al., 2019b; Cadene et al., 2019), visual grounding (Yu et al., 2018; Hong et al., 2019), and image captioning (Xu et al., 2015; You et al., 2016; Anderson et al., 2018; Lu et al., 2018; Huang et al., 2019).

Image captioning has many uses, including assisting visually impaired people in better perceiving their surroundings, extending human-robot interactions, and many others. In the image captioning task, the model generates a textual caption for a given input image. This task is simple for humans but difficult for computers since it includes detecting the main objects in an image, understanding their relationships to each other, and describing them in natural language.

In the past few years, most image captioning methods (Vinyals et al., 2015; Xu et al., 2015; Anderson et al., 2018; Lu et al., 2017) adopt an encoder-decoder framework, inspired by the sequence-to-sequence model for machine translation (Sutskever et al., 2014). Traditionally in such frameworks, a deep convolutional neural network (CNN) encoder converts the input image into region-based visual features, while a recurrent neural network (RNN) decoder produces caption words based on the extracted visual features. Recent captioning models tend to replace the RNN model in the decoder component with Transformer (Vaswani

et al., 2017) because of its potential for parallel training, richer representation, and better performance with long dependencies. The Transformer architecture, which records correlations between areas and words using self-attention, is an excellent example of such an encoder-decoder architecture (Herdade et al., 2019; Li et al., 2019; Yu et al., 2019a; Cornia et al., 2020; Mishra et al., 2021).

Despite these efforts, these methods and datasets are mostly restricted to 2D images and never utilize 3D information. In this paper, we are interested in exploiting the depth information of RGB-D scenes to improve the description generation task. Towards this goal, we propose a Transformer-based encoder-decoder model to generate multi-sentence descriptions for the 3D scenes. We aim to benefit from the existence of the depth map to enrich the image description and solve the lack of cognition on the spatial relationship in realistic 3D scenes. RGB-D data can be captured by low-cost depth sensors such as Microsoft Kinect, which extends traditional RGB recognition by integrating depth information. Depth cameras provide spatial data that can help in understanding the spatial layout of the scene's objects.

The depth and RGB images are separate modalities acquired by the sensor, and all of them provide us with information about how the scene appears. Depth images, unlike RGB images, do not offer fine visual details of the objects but rather present more distinct silhouettes of them. The depth channel has already been utilized successfully with CNNs to increase classification accuracy (Schwarz et al., 2015; Eitel et al., 2015) since it has complimentary information to the RGB channels and contains structural information of the image. In addition, depth images show a better description of the relationship between the 2D and 3D spatial positions of the objects in a scene, which enriches the scene description. Depth images can be captured by low-cost depth sensors such as Microsoft Kinect or can easily be acquired with an off-the-shelf pre-trained depth estimation model.

The growing availability of affordable 3D data acquisition devices and pre-trained depth estimation models with richer information in 3D data led us to investigate the best way for integrating depth (D) with color information (RGB) to get better captions and enhance the performance of the image captioning task. To demonstrate the effectiveness of our approach, we analyzed the NYU-v2 dataset. From our analysis, we discovered inconsistent labeling, such that the ground truth is not always represented, and the level of information in the descriptions ranges from a one-sentence general overview to a much more thorough one. So, we reviewed and cleaned it to make it consistent and focused. We trained the proposed framework on the cleaned version of the NYU-v2 dataset and the Stanford image paragraph captioning dataset. The Stanford image paragraph captioning dataset only contains RGB images. So, we get depth maps by using a pre-trained depth estimation model. That makes the proposed framework broadly applicable to any RGB captioning dataset. Extensive experiments show that our model can significantly improve 3D image captioning over the RGB-only baselines. In summary, our contributions are four-fold:

- We propose a cleaned version of the NYU-v2 dataset as a benchmark that is small in size and representative of both 2D and 3D relations.

- We propose an end-to-end framework for generating a multi-sentence description of a 3D scene.

- Besides the cleaned version of the NYU-v2 dataset, we also train the proposed model on the Stanford image paragraph captioning dataset with estimated depth maps, which makes the proposed framework applicable to any RGB image captioning dataset.

- Experimental results show the effectiveness of our method in exploiting depth information and generating better captions.

## 2 Related work

### 2.1 Multi-sentence image captioning

The majority of recent research on image captioning concentrates on single-sentence captions. However, providing a single statement is insufficient for expressing a semantically rich image. Krause et al. (2017) introduced the task of image paragraph captioning using an encoder-decoder framework and a hierarchical

recurrent neural network. In particular, a sentence RNN was used to produce sentence topics, while several word RNNs are in charge of producing words. Following that, Chatterjee & Schwing (2018) combined coherence vectors with global topic vectors to create various cohesive paragraph topics. Also, Wang et al. (2019) proposed a convolutional auto-encoding method to effectively utilize the image representation throughout the decoding process to extract valuable and representative topics over region-level features. To solve the drawbacks of these RNN-decoder-based approaches, Li et al. (2020) proposed a Dual-CNN language model for generating a paragraph that describes a given image.

## 2.2 3D Scene understanding

Several studies have investigated tasks for 3D scene understanding, such as text-to-image alignment (Kong et al., 2014) or generating referring expressions (Chen et al., 2020; Mauceri et al., 2019) for a specific object in an image. Regarding the caption generation task, Lin et al. (2015) proposed a framework for automatically generating multi-sentence linguistic descriptions of complicated indoor scenes. The primary goal of their work was to construct a 3D parsing system to produce a semantic representation of the scene. Because this method is hand-crafted, it rarely generates many sentences from a single image. After that, Moon & Lee (2018) studied the development of natural language descriptions from indoor scenes using graph convolution networks to extract local features from a 3D semantic graph map and long short-term memory (LSTM) to construct scene descriptions. However, graph convolutional networks typically require auxiliary models (e.g., visual relationship detection and attribute detection models) to generate the visual scene graph in the picture in the first place.

## 2.3 Self-supervised learning (SSL)

SSL receives supervisory signals directly from the data. Creating an auxiliary pre-text task for the model from the input data is the basic concept behind SSL; as the model completes the auxiliary task, it learns about the underlying structure of the data. For instance, it learns to predict any unseen part of the input from the seen part. Autoencoders (Vincent et al., 2008) instantiate this principle in vision by predicting the missing parts at the pixel or token level. Recently, masked autoencoders (MAE) (He et al., 2022; Bachmann et al., 2022), a class of autoencoders, have attracted attention in computer vision. As the name implies, a masked autoencoder learns representations by reconstructing the input's randomly masked patches.

Another approach of SSL that has many successful applications in computer vision is the contrastive learning approach (Hadsell et al., 2006). In contrastive learning, the model learns the image representation by contrasting the input examples. Specifically, the model will try to minimize the representation distance between similar images. Current methods, implemented with Siamese networks (Bromley et al., 1993; Bertinetto et al., 2016), produce two distinct augmentations of samples and feed them into the networks for contrastive learning. SimSiam (Chen & He, 2021) and BYOL (Grill et al., 2020) are two recent, well-known research that used Siamese architectures and were pioneers in removing negative samples from contrastive learning.

## 3 Methodology

This section describes our approach to building a Transformer-based encoder-decoder model for multi-sentence 3D scene captioning. We examined three alternative fusion approaches for fusing depth and RGB images: pixel-level fusion (Figure 1(a)), feature-level fusion (Figure 1(b)), and hybrid fusion (Figure 1(c)). There are various ways to carry out each approach. We conducted an extensive study with a large number of combinations (30+ architectures) to determine the best one. As Figure 1 shows, the main architectural components of our proposed model are: depth fusion module, feature extractor backbone, Transformer-based encoder block, and BERT decoder block. Our model takes as input both RGB and depth images that represent the 3D scene. The fusion step is responsible for combining the two input images in pixel or feature spaces. The feature extractor backbone generates the feature map for the input image, while the Transformer-based encoder generates a new representation of this feature map. Finally, we use BERT as our language model.

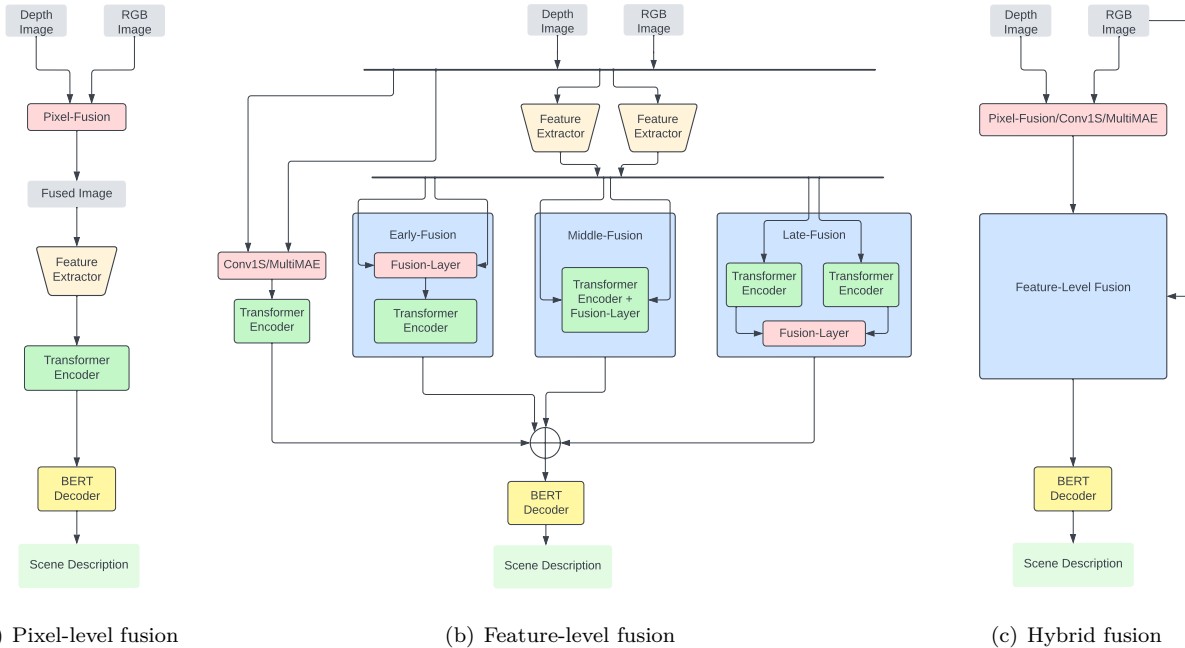

(a) Pixel-level fusion        (b) Feature-level fusion        (c) Hybrid fusion

Figure 1: Overview of our proposed model for a 3D scene description generation task. We studied three distinct approaches to fuse the depth map with the RGB image. (a) Apply fusion to a pixel space (pixel-level fusion). (b) Apply fusion in the feature space (feature-level fusion). (c) Use two feature fusion methods or combine pixel and feature fusion methods (Hybrid fusion)

## 3.1 Depth fusion

Fusing RGB and depth images can be performed in several ways. The different approaches used in our work can be generally divided into three classes: pixel-level fusion methods, feature-level fusion methods, and hybrid methods. Figure 1 represents an overview of these classes which will be explained in the following subsections.

### 3.1.1 Pixel-level fusion

In pixel-level image fusion, the original data from the source images containing complementary information of the same scene are merged to create a more informative image. We apply this type of fusion in three different ways, as illustrated in Figure 2:

1. Color and depth images are concatenated and mapped into a 3-channel input using the Distance maps fusion module (DMF) (Sofiiuk et al., 2020). It accepts as input a concatenation of a depth image and an RGB image. The DMF module processes the 4-channel input with 1 x 1 convolutions, followed by the ReLU activation function, and outputs a 3-channel tensor that can be sent into the RGB-trained backbone.

2. Inspired by Sofiiuk et al. (2021), we augment the weights of the first convolutional layer to make the pre-trained CNN model take 4-channel input rather than just an RGB image. We refer to this alteration of the network architecture by Conv1E.

3. Fusion at the level of HSV image representation as described by Mechal et al. (2022) by the following steps: First, map the RGB image to HSV representation. Then, replace the V channel with the depth map to get an HSD representation. Finally, use this HSD image as input to the pre-trained CNN model. An additional step conducts the inverse mapping from HSD to RGB before utilizing CNN. We refer to the output of this additional step as an RGBD image.

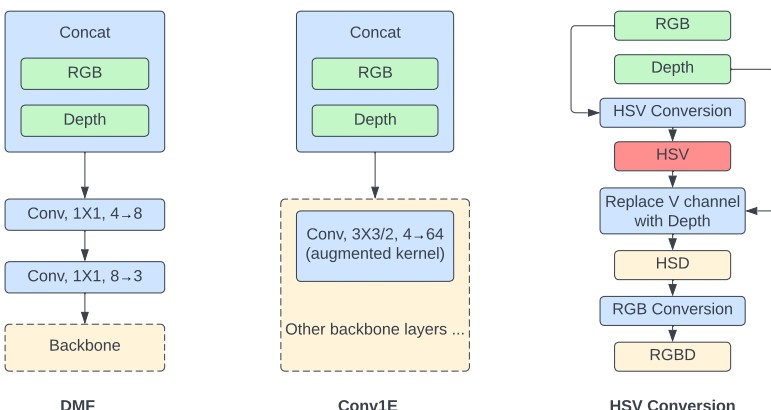

Figure 2: Different pixel-level fusion choices described in section 3.1.1

### 3.1.2 Feature-level fusion

Compared to pixel-level approaches, feature-level fusion methods process data at higher levels. Specifically, we apply feature extraction techniques first to generate a feature map for each image data source. Then, the fusion process deals with these feature maps and outputs a single feature map representing the different modalities. We studied four techniques to apply feature-fusion in our work: concatenation, cross-attention, Conv1S (Sofiiuk et al., 2021), and MultiMAE.

Regarding concatenation and cross-attention methods, we have three position choices to use in our model, illustrated in Figure 1(b). We denoted them by early-fusion, middle-fusion, and late-fusion. The **early-fusion** position is before the encoder block. We perform the fusion over the feature extractor outputs and right before the transformer encoder. The **middle-fusion** choice is to perform the fusion inside the encoder, briefly described in section 3.3. The last choice is **late-fusion**, where we pass each feature map to a separate encoder, then apply the fusion over the encoder outputs.

The idea of Conv1S, as shown in Figure 3(a), is to pass the additional input, e.g., depth image, to a convolutional block that outputs the same shape tensor as the first convolutional block in the backbone. The output of the first backbone convolutional layer is then element-wise summed with this tensor.

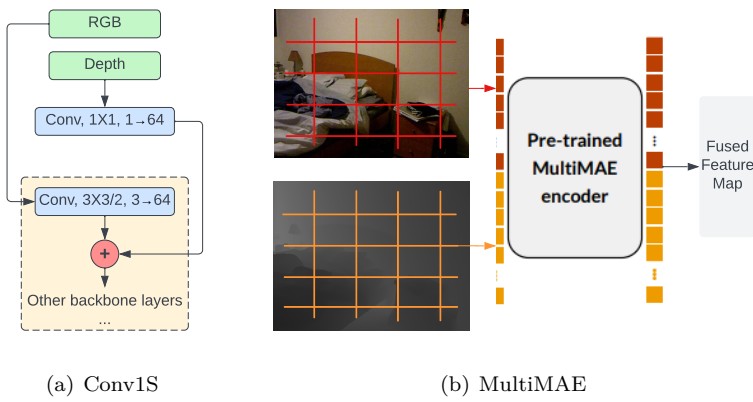

(a) Conv1S

(b) MultiMAE

Figure 3: Conv1S and MultiMAE methods. Example approaches for fusing RGB and depth images at the feature level

Another choice is using a pre-trained MultiMAE. Figure 3(b) shows that we pass the RGB and depth images to the pre-trained MultiMAE encoder and use the output feature map as the fused features to our model. In our work, we denote MultiMAE output features based on RGB and depth inputs by **MAE_CD**.

### 3.1.3 Hybrid fusion

Hybrid methods use the advantages of both pixel-level fusion and feature-level fusion methods by combining them (**pixel-feature**). An example of this approach is combining HSV conversion with feature-level fusion. Another way to apply hybrid fusion is by combining two feature-level fusion methods (**feature-feature**), as in applying the concatenation/cross-attention method with the MultiMAE features.

### 3.2 Feature extractor backbone

As the first step in our model, we need to extract features of input images. To do this, we employed a pre-trained feature extractor model, either a supervised model such as a pre-trained CNN (e.g., EfficientNet (Tan & Le, 2019)) or a pre-trained model in an unsupervised fashion such as a masked autoencoder (e.g., ViTMAE (He et al., 2022) and MultiMAE (Bachmann et al., 2022)). To investigate different methods to extract depth features, we trained the SimSiam (Chen & He, 2021) on the SUN RGB-D depth dataset (Song et al., 2015) and then used it as a depth feature extractor model. We did the same thing with the fused images (HSD/RGBD) and trained a SimSiam model on the fused images generated from the SUN RGB-D dataset.

### 3.3 Transformer-based encoder

Given a set of visual features extracted from the previous step, we passed them to a Transformer-based encoder. As shown in Figure 4, the transformer encoder consists of 5 layers that process the input consecutively: 1) normalization layer; 2) dense layer with ReLU; 3) dropout layer; 4) multi-head self-attention layer; 5) layer normalization with a residual connection. Figure 4 also shows the encoder in the case of the middle-fusion under the feature-level fusion methods. In this case, the encoder has two feature maps as inputs instead of one. Each feature map sequentially passes through a normalization, dense, and dropout

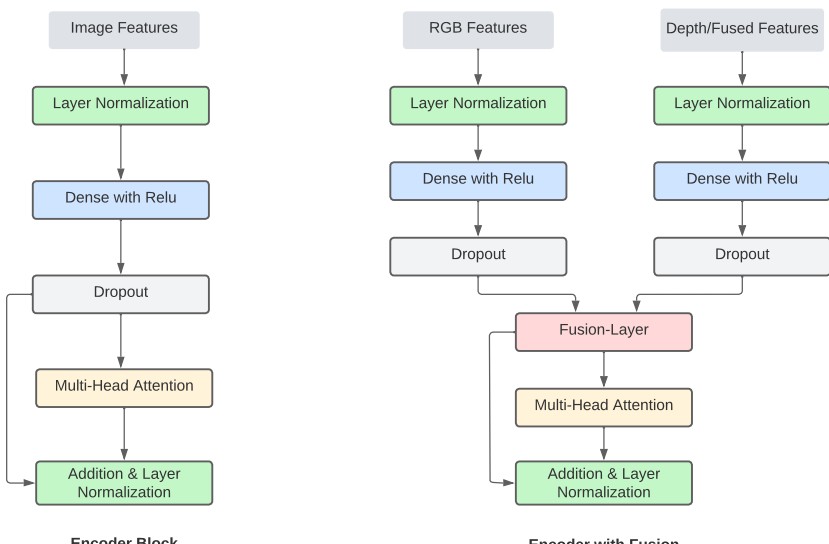

Figure 4: Transformer-based Encoder block architecture. **Left:** the base encoder block used in all experiments except in the case of the middle fusion. **Right:** the encoder block used in the case of the middle-fusion approach

layer. Then, the fusion layer, either concatenation or cross-attention, combines these projected features. After that, we apply the multi-head self-attention to the fused output. Finally, apply the layer normalization with a residual connection.

### 3.4 BERT decoder

The encoder outputs are fed into a language decoder model to generate scene descriptions. We use an uncased pre-trained BERT model (Devlin et al., 2018) as a language decoder. BERT is trained on English Wikipedia and BooksCorpus using a combination of Masked Language Modeling and next-sentence prediction tasks. We fine-tuned the BERT decoder during the training to steer the language model towards the new objective.

## 4 Datasets

### 4.1 NYU-v2 dataset

To test the proposed framework, we use the NYU-v2 dataset (Lin et al., 2015), which has 1449 RGB-D images of indoor scenes and one description per image. The number of sentences in each description ranges from one to ten, with an average of three. Following the partition employed in Lin et al. (2013), these images are split into a training set and a testing set. The training set includes 795 scenes, while the testing set includes the remaining 654.

**Dataset cleaning**

Table 1: Original dataset statistics. Shows how many times each object class is mentioned in the text in relation to all visual occurrences of that class

|  | mantel | counter | toilet | sink | bathtub | bed | headb. | table | shelf | cabinet | sofa | chair | chest | refrig. | oven | microw. | blinds | curtain | board | monitor | printer |
|---|---|---|---|---|---|---|---|---|---|---|---|---|---|---|---|---|---|---|---|---|---|
| % mentioned | 41.4 | 46.4 | 90.6 | 78.2 | 55.6 | 79.4 | 9.2 | 53.4 | 32.9 | 27.7 | 51.2 | 31.6 | 45.7 | 55.7 | 31.4 | 56.0 | 26.2 | 23.0 | 28.6 | 66.7 | 44.3 |

Because the dataset's annotators had no idea of the classes of interest in the images, the descriptions in the dataset are inconsistent. Moreover, the annotators were asked to describe the scene to someone who does not see it without any additional instructions. As a result, the descriptions vary from a general one-sentence summary to one that is more detailed. The dataset has 21 objects of interest classes, as mentioned in Lin et al. (2013). According to Kong et al. (2014), Table 1 shows the object of interest classes and the percentage of referring times to each object concerning the number of all visual objects of that class. This percentage, as we can see, ranges from 9% for items like a "headboard" to 91% for items such as a "toilet."

Another issue we discovered was the non-uniform point of reference while describing the scene. Sometimes it's from the viewpoint of the viewer, and other times it's from the perspective of the described object.

To address the inconsistency in the dataset descriptions, we manually reviewed the dataset and applied consistent relabeling. We follow the following rules in the relabelling process:

- Concentrate on the 21 objects of interest (listed in table 1).

- Describe the object's appearance, such as shape, color, and material.

- Describe the 2D and 3D spatial relationships between the scene's objects.

- Unite the point of reference to the viewer's point of view.

Figure 5 shows examples from the dataset before and after the relabelling process. In the first two images (5(a), 5(b)), for instance, the scene caption describes objects that are not part of the interest classes (e.g., fan, ironing board, lamp, and walls). In such cases, we remove the description of those objects to keep the focus on the interest classes. The next two images (5(c), 5(d)), on the other hand, represent a case where an object that belongs to the interest classes is not mentioned in the description (e.g., cabinets). In that

case, we add a description to this object. The example where we changed the point of reference to apply the final rule and unify the point of view to the viewer is shown in the images (5(e), 5(f), 5(g)). The final case illustrated in the last two images (5(h), 5(i)), where we keep the description as is because it satisfies the conditions. 1380 images were modified overall after cleaning, leaving 69 images that remained unchanged.

To summarize, as shown in Figure 5, there were 4 cases while cleaning the dataset:

1. We removed the description of objects that are not part of the interest classes (e.g., fan).

2. We added a description to the object that belongs to the interest classes if it's missing in the description (e.g., cabinets).

3. We changed the point of reference to unify it to the viewer.

4. No change happened to the description if it satisfies the conditions.

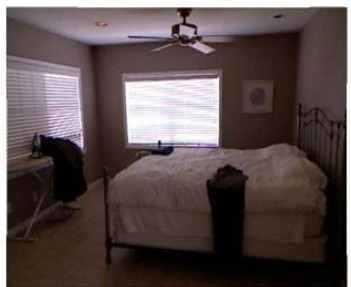

(a) **Before:** A bedroom with dark walls and a bed against the right wall. A fan is on the ceiling above the bed. There is an ironing board by the left window. Another window is in the back wall.
**After:** A bed with metal frame is against the right wall in this bedroom. White blinds cover two windows, one behind the bed and the other to its left.

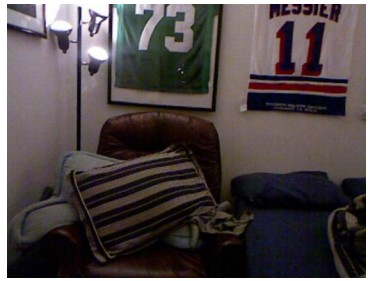

(b) **Before:** A room with a leather chair in the corner, on which two pillows are placed. Behind the chair there is a tall lamp with three light bulbs. There are three sports t-shirts hanging on the wall behind the chair.
**After:** It is a room with a leather chair in the corner, on which two pillows are placed.

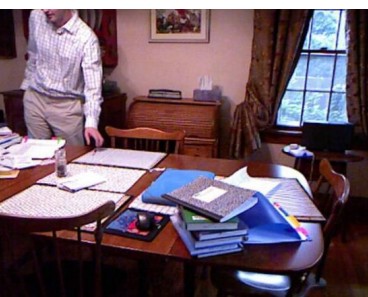

(c) **Before:** A tall man is working in his study room. He is standing next to a table which is covered with paper and notebooks. There are three chairs next to the table.
**After:** There is a table covered with paper and notebooks in front of us. Three chairs are around the table. Behind the table are a small cabinet and a window curtain to its right.

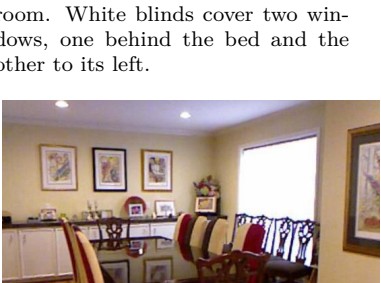

(d) **Before:** This is a dining room with a long wooden table and white and red chairs next to the table. It could also be a meeting room since there are extra chairs behind the table. The room has four paintings on the walls.
**After:** This is a dining room with a long wooden table and white and red chairs next to the table. It could also be a meeting room since there are extra chairs behind the table. By the back wall are white cabinets.

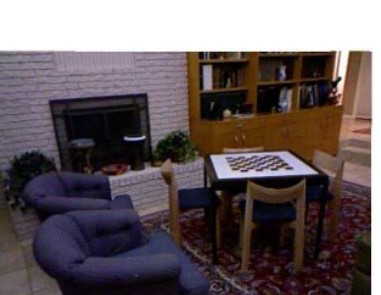

(e) **Before:** A living room with a chess table in the middle of the room. To the right of the table are two blue armchairs. Behind the table and armchairs is a fireplace. Along the same wall is a cabinet with books.
**After:** It's a living room with a chess table surrounded by four small chairs in the middle of the room. Two blue armchairs are to the left of the table. Behind the table and armchairs is a fireplace. Along the same wall is a cabinet with books.

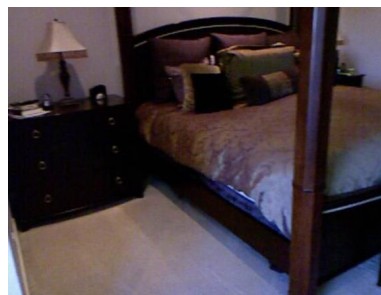

(f) **Before:** A neatly furnished room with a double bed, pillows on top of the bed and a cabinet with six drawers to the right of the bed. On top of the cabinet is a lamp.
**After:** This is a neatly furnished bedroom. There is a double bed with a dark wooden frame and pillows on top of the bed. A cabinet with drawers is to the left of the bed.

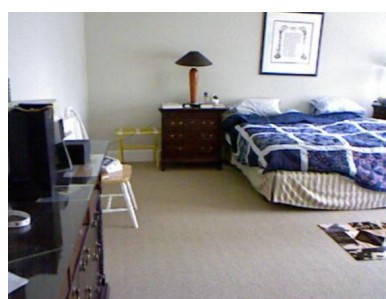 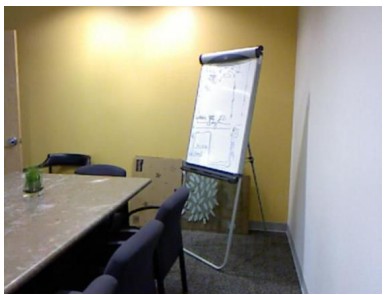 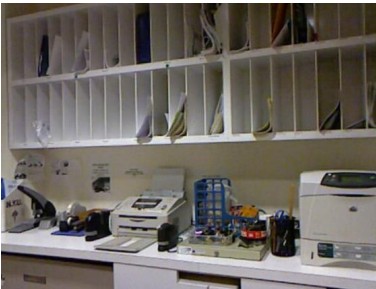

(g) **Before:** This is a bedroom with a bed that has two pillows on top, a blue-white bed sheet, two night lamps on both sides of the bed, and a cabinet directly on our right.
**After:** This is a large bedroom with a bed that has two pillows on top and a blue-white bed sheet. There is a night-stand on each side of the bed. Near the left corner is a small, yellow table or chair. A cabinet is directly on our left and a chair behind it.

(h) **Before:** This is a picture of a meeting room. There are purple chairs around the meeting table, and a white board with drawings.
**After:** This is a picture of a meeting room. There are purple chairs around the meeting table, and a white board with drawings.

(i) **Before:** Two printers are on top of a cabinet. There are also a bunch of other things on top of the cabinet. Above the printers there is a mail holder.
**After:** Two printers are on top of a cabinet. There are also a bunch of other things on top of the cabinet. Above the printers there is a mail holder.

Figure 5: Examples from the NYU-v2 dataset before and after the cleaning step. **(a) and (b)** are examples of removing the description of objects other than the classes of interest. **(c) and (d)** are examples of adding descriptions to objects belonging to interest classes. **(e), (f), and (g)** are examples of change the point of reference. **(h) and (i)** are examples of not changing the scene description

## 4.2 Stanford image paragraph dataset

The Stanford image paragraph dataset (Krause et al., 2017) is the benchmark dataset for RGB image paragraph captioning. It was collected from MS COCO (Lin et al., 2014) and Visual Genome (Krishna et al., 2017) datasets. This dataset contains 14575 training images, 2487 validation images, and 2489 testing images, each with a human-annotated paragraph. Each paragraph has an average of 67.5 words, 5.7 sentences, and 11.91 words per sentence. Since the dataset only contains RGB images, we used a DPT-Hybrid (Ranftl et al., 2021) trained on Omnidata (Eftekhar et al., 2021) for depth estimation.

# 5 Experiments

## 5.1 Evaluation metrics

We evaluate the caption quality using the standard automatic evaluation metrics, namely BLEU-1, BLEU-4 (Papineni et al., 2002), ROUGE-1, ROUGE-2, ROUGE-L (ROUGE, 2004), METEOR (Denkowski & Lavie, 2014), and CIDEr (Vedantam et al., 2015). They are denoted as B-1, B-4, R-1, R-2, R-L, M, and C, respectively. METEOR and ROUGE-L are f-measure scores, ROUGE-N is a recall-based measure, and BLEU is a precision-based measure. METEOR has more flexible matching requirements. CIDER is a consensus-based captioning measure that weights n-grams across all references using term frequency inverse document frequency (TF-IDF).

## 5.2 Baselines

To answer the central question in our study, "*Can we use the depth information to produce better captions?*" we evaluate our proposed model with different depth fusion approaches against previous work and an ablated baseline. Our main goal is not to achieve state-of-the-art results but to demonstrate that the proposed approach benefits from the depth signal to improve image captioning performance.

**The baseline model** takes as input the RGB image, uses an EfficientNet pre-trained backbone to extract the image features, and then feeds the image features to the Transformer-based encoder. After that, the language decoder (BERT) uses the encoder output to generate the caption. In one experiment, we trained the BERT from scratch, whereas in the other, we fine-tuned the pre-trained BERT.

**Show and Tell** (Xu et al., 2015) use RNN as a textual decoder with attention over the image features to produce the related words.

**Show and Tell with transformers**, a modified version of Xu et al. (2015), uses a 3-layer Transformer-decoder. We use the implementation provided by Tensorflow (2022).

**VirTex**(Desai & Johnson, 2021) jointly train a convolutional network (ResNet-50) and Transformers (two unidirectional Transformers) from scratch.

### 5.3 Implementation details

We trained our model end-to-end with the decoder initialized by the pre-trained BERT-BASE model. The input images are resized to $224 \times 224$ resolution. We keep the pre-trained feature extractor backbone frozen across all experiments. The output dimension of the token embedding layers in the BERT decoder is 768, and the input visual features are also mapped into 768 with a linear projection inside the transformer encoder. We used a grid search to select the best hyperparameters in our study. We adjusted the number of attention heads, learning rate, BERT dropout, and encoder dropout separately for each experiment. The whole model is trained with cross-entropy loss. We use a batch size of 16 through all experiments unless explicitly stated otherwise. For optimization, we use AdamW optimizer(Kingma & Ba, 2014). We use the Tensorflow (Abadi et al., 2016) framework and run all experiments on a single Nvidia Geforce RTX 2070 Super GPU.

## 6 Results and discussion

### 6.1 Main results

#### 6.1.1 With ground truth depth - NYUv2

**2D vs. 3D Captioning.** We compare our 3D captioning approach to the baseline models on the official test split of the NYUv2 sentence dataset (Lin et al., 2013) in Table 2. We report BLEU-1, BLEU-4, ROUGE-L, and METEOR results. Regarding our approach, we list the results of the best configuration for each depth fusion method. We fine-tuned the hyperparameters of each method independently to ensure a fair comparison. Table 2 shows that our approach outperforms the baselines with a remarkable margin. The hybrid fusion method achieves the highest score for the METEOR metric (with a **3.59** improvement), while the feature-level fusion method is better in BLEU-1, BLEU-4, and CIDEr metrics (with **3.58**, **1.65**, and **8.9** improvements, respectively). The ROUGE scores for the two methods are comparable (**1.17** and **1.20** for ROUGE-1 and ROUGE-L, respectively).

Table 2: Comparison of 3D captioning results obtained by our model and the baseline models. **Black** indicates results better than baselines and **Blue** indicates the best results

| | Modality | Inputs | Method | B-1 | B-4 | R-1 | R-L | M | C |
|---|---|---|---|---|---|---|---|---|---|
| Show-and-Tell (Xu et al., 2015) | 2D | RGB | - | 26.22 | 6.54 | 40.41 | 32.65 | 23.61 | 10 |
| Show-and-Tell(Transformer) (Tensorflow, 2022) | 2D | RGB | - | 35.81 | 6.95 | 35.14 | 29.3 | 24.95 | 13.01 |
| VirTex (Desai & Johnson, 2021) | 2D | RGB | - | 30.56 | 6.39 | 39.72 | 36.08 | 25.51 | - |
| Our baseline(BERT from scratch) | 2D | RGB | - | 35.58 | 8.67 | 42.57 | 38.76 | 26.24 | - |
| Our baseline(pre-trained BERT) | 2D | RGB | - | 42.4 | 10.99 | 44.5 | 41.04 | 30.25 | 17.54 |
| Ours (Pixel-level) | 3D | RGB+Depth | DMF | **44.69** | **11.65** | 43.67 | 40.28 | **30.83** | **19.2** |
| Ours (Feature-level) | 3D | RGB+Depth | Late-fusion (Cross-attention) | **45.98** | **12.64** | **45.67** | **42.22** | **32.87** | **26.44** |
| Ours (Hybrid) | 3D | RGB+MAE_CD | Late-fusion (Concat) | 41.05 | 10.62 | **45.65** | **42.24** | **33.84** | 22.36 |

**Pixel-Level fusion.** Table 3 compares the performance of the three pixel-level fusion methods. DMF performs the best in all assessed metrics except for METEOR. The highest METEOR score is achieved by HSV conversion with RGBD.

Table 3: Evaluation results of the pixel-level fusion methods described in section 3.1.1

| Method | Inputs | B-1 | B-4 | R-L | M |
|---|---|---|---|---|---|
| DMF | RGB+Depth | **44.69** | **11.65** | **40.28** | 30.83 |
| Conv1E | RGB+Depth | 43.5 | 10.86 | 40.23 | 30.85 |
| HSV Conversion | HSD | 40.69 | 9.44 | 37.16 | 28.17 |
|  | RGBD | 43.64 | 11.02 | 39.98 | **32.04** |

**Featur-Level Fusion.** We examine the effectiveness of the feature fusion methods presented in Section 3.1.2 in Table 4. This table also shows the positions where fusion can occur. Overall, all of the methods outperform the baseline models. Using a cross-attention layer to perform late fusion RGB and depth images achieves the best results in all assessed metrics.

Table 4: Evaluation results of the feature-level fusion methods described in section 3.1.2

| Method | Inputs | B-1 | B-4 | R-L | M |
|---|---|---|---|---|---|
| Conv1S |  | 44.47 | 11.41 | 40.48 | 31.04 |
| MultiMAE |  | 43.64 | 11.02 | 39.98 | 32.04 |
| Early-Fusion(Cross-attention) | RGB+Depth | 42.57 | 11.4 | 41.76 | 31.65 |
| Middle-Fusion(Cross-attention) |  | 43.72 | 11.83 | 40.73 | 31.05 |
| Late-Fusion(Cross-attention) |  | **45.98** | **12.64** | **42.22** | **32.87** |

**Concatenation vs. Cross-attention** Table 5 compares concatenation and cross-attention as a used fusion layer. We first evaluate in case of the early-fusion position, and the results for cross-attention outperform the concatenation method. We then evaluated the results of the middle-fusion position, and concatenation is better than cross-attention. Lastly, we evaluated the results of the late-fusion position with RGB and depth images as inputs, as well as RGB and MultiMAE features as inputs. Cross-attention outperforms concatenation on RGB and depth images. While for RGB and MultiMAE features, cross-attention results are better in the BLEU score, concatenation results are better in both ROUGE and METEOR scores.

Table 5: Comparison of concatenation and cross-attention methods at three different positions of our model. **Bold** indicates the best results in each position

| Inputs | Position | Method | B-1 | B-4 | R-L | M |
|---|---|---|---|---|---|---|
| RGB+Depth | Early-fusion | Concat | 41.27 | 10.86 | 41.36 | 30.8 |
|  |  | Cross-attention | **42.57** | **11.4** | **41.76** | **31.65** |
| RGB+Depth | Middle-fusion | Concat | **45.02** | **12.31** | **41.22** | **31.79** |
|  |  | Cross-attention | 43.72 | 11.83 | 40.73 | 31.05 |
| RGB+Depth | Late-fusion | Concat | 44.36 | 11.29 | 40.83 | 31.05 |
|  |  | Cross-attention | **45.98** | **12.64** | **42.22** | **32.87** |
| RGB+MAE_CD | Late-fusion | Concat | 41.05 | 10.62 | **42.24** | **33.84** |
|  |  | Cross-attention | **45.86** | **11.63** | 40.64 | 32.25 |

**Hybrid Fusion.** In Table 6, we compare the performance of the hybrid fusion approach. The method can be implemented by combining pixel and feature techniques (pixel-feature) or by combining two feature methods (feature-feature) (Section 3.1.3). The best results for BLEU-4 and ROUGE-L (**1.76** and **1.23** improvements, respectively, over the baseline) are obtained with pixel-feature. The best results for BLEU-1 and METEOR (**4.54** and **3.59** improvements, respectively, over the baseline) are obtained with feature-feature.

Table 6: Evaluation results of hybrid fusion approach. **Top:** evaluation results of pixel and feature fusion method combination (pixel-feature). **Bottom:** evaluation results for fusing two feature-based methods (feature-feature). **Black** indicates the best results in each category and **Blue** indicates the best results overall

|  | Method | Inputs | B-1 | B-4 | R-L | M |
|---|---|---|---|---|---|---|
| Pixel-Feature | Early-fusion(Cross-attention) | RGB+RGBD | 42.59 | 11.38 | 42 | 31.61 |
|  | Middle-fusion(Concat) | RGB+RGBD | 43.88 | 11.89 | 41.45 | 31.95 |
|  | Late-fusion(Cross-attention) | RGB+RGBD | **45.97** | **12.75** | **42.27** | **32.71** |
| Feature-Feature | Conv1S,Middle-fusion(Concat) | RGB+Depth+MAE_CD | 44.31 | 10.6 | 39.63 | 31.27 |
|  | Middle-fusion(Concat) | RGB+MAE_CD | **46.94** | 12.25 | 41.68 | 32.16 |
|  | Late-fusion(Concat) | RGB+MAE_CD | 41.05 | 10.62 | **42.24** | **33.84** |
|  | Late-fusion(Cross-attention) | RGB+MAE_CD | 45.86 | **11.63** | 40.64 | 32.25 |

**Fusion Position.** We compare the performance of our model with the different fusion position choices as described in section 3.1.2. Table 7 demonstrates that when employing the cross-attention as a fusion approach, the late-fusion position works better than both early and middle fusion. Middle-fusion works equally well or better than late-fusion when employing the concatenation approach. When the inputs are RGB and MultiMAE_CD, middle-fusion performs better for BLEU scores, while late-fusion outperforms in the ROUGE and METEOR scores.

Table 7: comparing the results of our model performance with the different fusion positions (early-fusion, middle-fusion, and late-fusion). **Black** indicates the best results among the three positions with respect to the input and **Blue** indicates the best results overall

| Fusion Position | Inputs | Method | B-1 | B-4 | R-L | M |
|---|---|---|---|---|---|---|
| Early-Fusion |  |  | 41.27 | 10.86 | **41.36** | 30.8 |
| Middle-Fusion | RGB+Depth | Concat | **45.02** | **12.31** | 41.22 | **31.79** |
| Late-Fusion |  |  | 44.36 | 11.29 | 40.83 | 31.05 |
| Early-Fusion |  |  | 42.57 | 11.4 | 41.76 | 31.65 |
| Middle-Fusion | RGB+Depth | Cross-attention | 43.72 | 11.83 | 40.73 | 31.05 |
| Late-Fusion |  |  | **45.98** | **12.64** | **42.22** | **32.87** |
| Early-Fusion |  |  | 37.55 | 10.26 | 40.09 | 28.98 |
| Middle-Fusion | RGB+RGBD | Concat | 43.88 | **11.89** | **41.45** | **31.95** |
| Late-Fusion |  |  | **45.75** | 11.57 | 41.1 | 31.66 |
| Early-Fusion |  |  | 42.59 | 11.38 | 42 | 31.61 |
| Middle-Fusion | RGB+RGBD | Cross-attention | 38.91 | 9.43 | 39.35 | 28.27 |
| Late-Fusion |  |  | **45.97** | **12.75** | **42.27** | **32.71** |
| Early-Fusion |  |  | 42.77 | 11.26 | 42.01 | 31.53 |
| Middle-Fusion | RGB+HSD | Cross-attention | 40.44 | 10.47 | 40.09 | 29.55 |
| Late-Fusion |  |  | **45.85** | **12.65** | **42.11** | **32.66** |
| Middle-Fusion | RGB+MAE_CD | Concat | **46.94** | **12.25** | 41.68 | 32.16 |
| Late-Fusion |  |  | 41.05 | 10.62 | **42.24** | **33.84** |

### 6.1.2 With predicted depth

**Stanford image paragraph.** Besides the experiments on the NYUv2 dataset, we compare our 3D captioning approach to the 2D captioning baseline model on the Stanford image paragraph captioning dataset in Table 8. We report BLEU-4, ROUGE-1, ROUGE-2, ROUGE-L, METEOR, and CIDEr scores. Table 8 shows that, even with estimated depth maps, our approach outperforms the baseline model with a notable margin, particularly for ROUGE and CIDEr scores. The best results are achieved using the feature-level fusion method with cross-attention, which improves ROUGE-1 by **2.12**, ROUGE-2 by **1.86**, ROUGE-L by **0.71**, and CIDEr by **3.01** points.

Table 8: Comparison of 3D captioning results obtained by our model and the baseline model on the Stanford image paragraph captioning dataset. **Black** indicates results better than baseline and **Blue** indicates the best results

|  | Inputs | Method | B-4 | R-1 | R-2 | R-L | M | C |
|---|---|---|---|---|---|---|---|---|
| Region-Hierarchical (Krause et al., 2017) | RGB | - | 8.69 | - | - | - | 15.95 | 13.52 |
| Our 2D baseline(pre-trained BERT) | RGB | - | 8.53 | 35.77 | 11.31 | 27.56 | 13.89 | 13.6 |
| Ours 3D (Pixel-level) | RGB+Depth | DMF | 7.71 | **36.67** | **12.32** | 27.46 | 13.14 | **15.47** |
| Ours 3D (Feature-level) | RGB+Depth | Late-fusion (Cross-attention) | **8.66** | **37.89** | **13.17** | **28.27** | 13.74 | **16.61** |
| Ours 3D (Hybrid) | RGB+MAE_CD | Late-fusion (Concat) | 8.39 | **36.71** | **12.56** | **27.81** | 13.4 | **14.74** |

**NYU-v2.** Table 9 compares our 3D captioning approach with a 2D captioning baseline model on the NYU-v2 dataset using predicted depth instead of ground truth depth. The results confirm our previous finding on the Stanford image paragraph dataset that our approach outperforms the baseline model even when using estimated depth maps. The feature-level fusion approach with cross-attention achieves the best results in BLEU-4 (with a **1.55** improvement) and CIDEr (with a **7.99** improvement). The best METEOR score is obtained by the hybrid approach (with a **2.77** improvement).

Table 9: Comparison of 3D captioning results obtained by our model and the baseline model on the NYU-v2 cleaned dataset with predicted depth instead of ground truth depth. **Black** indicates results better than baseline and **Blue** indicates the best results

|  | Inputs | Method | B-4 | R-1 | R-2 | R-L | M | C |
|---|---|---|---|---|---|---|---|---|
| Our 2D baseline(pre-trained BERT) | RGB | - | 10.99 | 44.5 | 19.65 | 41.04 | 30.25 | 17.54 |
| Ours 3D (Pixel-level) | RGB+Depth | DMF | 10.24 | 41.46 | 17.57 | 38.49 | 29.46 | **18.73** |
| Ours 3D (Feature-level) | RGB+Depth | Late-fusion (Cross-attention) | **12.54** | **45.03** | **19.83** | **41.72** | **32.27** | **25.53** |
| Ours 3D (Hybrid) | RGB+MAE_CD | Late-fusion (Concat) | 10.79 | **45.16** | 18.98 | **41.24** | **32.95** | **23.03** |

### 6.1.3 NYU-v2 dataset cleaning

In Table 10, we study the dataset-cleaning effect. We evaluate the show-and-tell model, our baseline, and proposed models on the dataset before and after the cleaning step. Noticeably, the evaluation results of all models on the cleaned dataset outperform their results on the original dataset by a remarkable margin. This shows that the cleaned dataset can benefit not just our model but also other models.

Moreover, in the case of depth fusion, the proposed models perform worse than the baseline model on the original dataset. This indicates that the original dataset's inaccurate labeling has an impact on the depth signal's capturing and prevents the model from learning from the depth data.

Table 10: Results of our model and the show-and-tell model before and after the relabeling process

| Model | Dataset | Inputs | B-1 | B-4 | R-L | M |
|---|---|---|---|---|---|---|
| Show-and-Tell(Transformer) | Original | RGB | 33.37 | 5.14 | 26.38 | 22.85 |
| Ours | Original | RGB | 41.95 | 9.78 | 38.28 | 28.33 |
|  |  | RGB+Depth | 41.73 | 9.22 | 37.11 | 27.68 |
|  |  | RGB+MAE_CD | 39.8 | 7.44 | 35.18 | 26.07 |
| Show-and-Tell(Transformer) | Cleaned | RGB | 35.81 | 6.95 | 29.3 | 24.95 |
| Ours | Cleaned | RGB | 42.4 | 10.99 | 41.04 | 30.25 |
|  |  | RGB+Depth | **45.98** | **12.64** | 42.22 | 32.87 |
|  |  | RGB+MAE_CD | 41.05 | 10.62 | **42.24** | **33.84** |

### 6.2 Ablation study and analysis

**Early-fusion vs. Late-fusion.** The late-fusion approach has more parameters than the early-fusion approach because it uses two separate Transformer-based encoders compared to the early-fusion approach's single Transformer-based encoder. For a fair comparison, we compensate the number of parameters in both approaches by increasing the number of trainable variables in the early-fusion approach. We did that by simply using two encoders instead of one, where the output of the first encoder goes as input to the second.

Table 11 shows that, even after increasing the number of trainable variables in the early-fusion approach, the late-fusion approach still outperforms it. This implies the improvement comes from the fusion position, not the number of trainable variables.

Table 11: Evaluation results of early-fusion and late-fusion position after equalizing the number of learnable parameters

| Method | Params. | B-1 | B-4 | R-L | M |
|---|---|---|---|---|---|
| Early-Fusion(Cross-attention) | 344 | 42.57 | 11.4 | 41.76 | 31.65 |
| Early-Fusion(Cross-attention) | 358 | **46.07** | 12.43 | 41.49 | 31.88 |
| Late-Fusion(Cross-attention) | 358 | 45.98 | **12.64** | **42.22** | **32.87** |

**MultiMAE: RGB, Depth, or RGB+Depth.** Because MultiMAE was pre-trained on RGB, depth, and semantic segmentation, it may accept any of those modalities as input. In our work, we have RGB and depth images. So we can pass RGB, depth, or RGB+Depth to the pre-trained MultiMAE. As shown in the results of the previous experiments, using MultiMAE features based on RGB and depth modalities outperforms the baseline. *Is this improvement because of fusing depth modality with RGB or a better RGB representation by the MultiMAE?* In this ablation, we aim to answer this question by comparing the performance of our model with RGB, depth, and RGB+Depth MultiMAE features. We did all experiments in Table 12 by applying late-fusion using the concatenation method between the RGB image and the MultiMAE output features.

The results show that our model performs better with RGB+Depth MultiMAE features. Moreover, interestingly, using RGB-only MultiMAE features has comparable results as our baseline model. In conclusion, the findings in this section demonstrate that the MultiMAE features' ability to enhance performance is not due to a richer RGB representation but rather to the use of depth information.

Table 12: Ablation results of different MultiMAE input modalities. All experiments were carried out using late-fusion with the concatenation method between the RGB image and the MultiMAE output features

| Method | MultiMAE Inputs | B-1 | B-4 | R-L | M |
|---|---|---|---|---|---|
| Our baseline | - | 42.4 | 10.99 | 41.04 | 30.25 |
| | RGB | **44.5** | **10.83** | 39.5 | 30.2 |
| Late-Fusion(Concat) | Depth | 41.51 | 10.52 | 41.09 | 32.27 |
| | RGB+Depth | 41.05 | 10.62 | **42.24** | **33.84** |

**Feature extractor backbone.** Our experiments thus far are based on a frozen feature extractor backbone. In this ablation, we examine whether fine-tuning the feature extractor backbone will improve our results. To do this, we trained the entire model across several epochs with the frozen feature extractor backbone. Then, unfreeze a few of the feature extractor backbone's top layers and jointly train these last layers with the rest of our model.

Table 13 shows the results. We attempted to fine-tune a different number of top layers in the backbone. We did all experiments in Table 13 by applying early-fusion using the concatenation method between RGB and depth images. The results clearly suggest that our model performs better with a frozen backbone.

Table 13: Ablation results between frozen and fine-tuned feature extractor backbone. All experiments were carried out using early-fusion with the concatenation method between the RGB image and the depth images

| Backbone | Unfreezed-Layers | Inputs | B-1 | B-4 | R-L | M |
|---|---|---|---|---|---|---|
| Frozen | - | | **41.27** | **10.86** | **41.36** | **30.8** |
| Fine-tuned | last 40 | RGB+Depth | 38.18 | 9.93 | 40.35 | 28.72 |
| Fine-tuned | last 90 | | 38.17 | 9.83 | 40.14 | 28.59 |
| Fine-tuned | last 140 | | 38.5 | 10.13 | 40.59 | 29.11 |

## 7 Conclusion

In our work, we proposed a Transformer-based encoder-decoder model that can efficiently exploit depth information, resulting in improved image captioning. Our model takes the RGB and depth images representing the scene and fuses them to get richer information about the input scene. Moreover, we explore different fusion approaches to fuse the depth map with the RGB image. Extensive experiments on a cleaned version of the challenging NYU-v2 sentence dataset and the Stanford image paragraph captioning dataset validate the effectiveness and superiority of our approach. The experiments showed the capabilities of our framework and demonstrated the fusion of depth information can result in notable gains in image captioning, either as ground truth or estimated. Cleaning the NYU-v2 dataset has been shown to impact the model's performance significantly. Overall, we are hopeful that our cleaned version of the NYU-v2 dataset and framework can support further study in the area of 3D visual language.

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
