# OpenReview forum: "Enhancing image captioning with depth information using a Transformer-based framework"
_TMLR — Rejected by TMLR_

### Review · Reviewer_mNHc · 2023-01-04

**Summary Of Contributions:**

This paper studies how to improve image captioning using depth information encoded by depth maps. In particular, the authors 1) conduct a systematic study on the model architecture to find the best way to combine depth and RGB information, and 2) cleanup existing NYU-v2 dataset to better fit the depth-based image captioning problem. The authors conduct extension experiments and analysis to study the effect of model architecture and the importance of data cleanup.

**Audience:**

Yes

**Broader Impact Concerns:**

There's not additional concern of this work.

**Claims And Evidence:**

Yes

**Requested Changes:**

1. The authors should provide solid justifications for the proposed task, e.g. being essential for some applications, enable fundamental vision research, etc.
2. Explain the physical meaning and justify the proposed data cleaning. Furthermore, the authors should empirically show that it indeed addresses issues of existing datasets.
3. Verify the findings are generalizable rather than simply overfit the dataset.

**Strengths And Weaknesses:**

Strength
* The authors conduct extensive experiment to study the effect of model architecture and validate the superior performance of the proposed model
* The authors clearly demonstrate that existing NYU-v2 dataset may not be suitable for the target task and that additional data pre-processing is necessary
* The authors introduce a new dataset based on NYU-v2 for future image captioning research

Weakness
* It is not clear why the target task and the proposed method are important. The authors essentially define a new task by modifying the NYU-v2 dataset, but the paper does not explain why the new task is meaningful and important. Without further explanation, it's hard to justify the significance of the contribution.
* While the authors show that data cleaning is important for the target task, the paper does not explain what the problem of the original NYU-v2 dataset is and what the implication and physical meaning of the data cleaning is. Based on the paper, it's hard to judge whether the data processing really fix known issues of NYU-v2 or not.
* While the authors conduct extensive study on the model architecture, the significance of the finding is not clear. In particular, it is not clear whether the findings and conclusions can generalize to other datasets or tasks.

---

> ### Author Response · Authors · 2023-03-16
> **Response to Reviewer mNHc**
>
> Thank you for your comprehensive review and valuable feedback! We address your comments one by one as follows:
>
> ### 1. Provide solid justifications for the proposed task
>
> We want to clarify that [1] and [2] previously addressed the image captioning task based on RGB-D data. More details about their works are mentioned in subsection 2.2.
>
> Regarding the importance of the task, RGB-D data can be captured by low-cost depth sensors such as Microsoft Kinect or can easily be acquired with an off-the-shelf pre-trained depth estimation model. In either case, traditional RGB image is extended by integrating depth information which provides spatial data that can help in understanding the spatial layout of the scene's objects.
>
> The growing availability of affordable 3D data acquisition devices and pre-trained depth estimation models with richer information in 3D data led us to investigate the best way for integrating depth (D) with color information (RGB) to get better captions and enhance the performance of the image captioning task. We improved the discussion on the importance of the task in the introduction section of the paper.
>
> ---
>
> ### 2. NYU-v2 dataset cleaning
>
> Thanks for this comment! Kindly refer to the detailed response for this point in the [common reply (part 2 - point 2)](https://openreview.net/forum?id=PtrK8Aoe2M&noteId=tql28zYAMs)  under the '**NYU-v2 dataset cleaning**' title.
>
> ---
>
> ### 3. The concern about overfitting the dataset
>
> Thanks for this comment! Kindly refer to the detailed response for this point in the [common reply (part 1 - point 1)](https://openreview.net/forum?id=PtrK8Aoe2M&noteId=j354ozGi7U) under the ‘**Test the proposed approaches using predicted depth only**' title.
>
> Please let us know if you have any further questions or suggestions.
>
> ---
> [1] Lin, Dahua, et al. "Generating multi-sentence lingual descriptions of indoor scenes." arXiv preprint arXiv:1503.00064 (2015). \
> [2] Moon, Jiyoun, and Beomhee Lee. "Scene understanding using natural language description b

---

### Review · Reviewer_qUcw · 2023-02-27

**Summary Of Contributions:**

This paper focuses on the problem of image captioning. The author proposes enhancing image captioning models' performance with both RBG images and related depth maps. The paper explores different transformer-based structures to fuse the information from the RGB and depth branches. The experimental evaluations are conducted on the NYU-v2 dataset, and the results demonstrate that combining both RGB and depth information can improve the performance of image captioning.

**Audience:**

Yes

**Broader Impact Concerns:**

No ethical issues are involved.

**Claims And Evidence:**

No

**Requested Changes:**

The proposed topic is interesting and worse further in-depth investigation. The authors are suggested to do an in-depth analysis, and then design and develop new and inspirational architectures and algorithms for solving the problem. Besides, more experimental evaluations and convincing results are needed to support the claims, and thus can make the manuscript stronger.

**Strengths And Weaknesses:**

Strengths:
1. The paper writing is very clear and easy to follow.
2. The topic of combining RBG images and depth maps to improve the performance of image captioning is interesting and worth investigating.

Weaknesses:
1. Though the topic is interesting, the proposed solution is very simple and straightforward. The technical contribution is very limited as all the counterparts exist or are proposed by previous works. I would instead treat the manuscript as an experimental report, e.g., exploring the effectiveness of all existing solutions on a specific dataset and task.
2. The experimental evaluations are also not that convincing. Only one dataset NYU-v2 is tested. More convincing results and datasets are needed to validate the effectiveness of different design details.

---

> ### Author Response · Authors · 2023-03-16
> **Response to Reviewer qUcw**
>
> Thank you for your comprehensive review and valuable feedback! We address your comments one by one as follows:
>
> ### 1. Limited technical contribution
>
> Thanks for this comment! Kindly refer to the detailed response for this point in the [common reply (part 2 - point 3)](https://openreview.net/forum?id=PtrK8Aoe2M&noteId=tql28zYAMs) under the ‘**Limited technical contribution**' title.
>
> ---
>
> ### 2. The concern about overfitting the dataset
>
> Thanks for this comment! Kindly refer to the detailed response for this point in the [common reply (part 1 - point 1)](https://openreview.net/forum?id=PtrK8Aoe2M&noteId=j354ozGi7U) under the ‘Test the proposed approaches using predicted depth only' title.
>
> Please let us know if you have any further questions or suggestions.

---

### Review · Reviewer_thm9 · 2023-03-05

**Summary Of Contributions:**

The paper studies whether depth input along with RGB input can help the task of image captioning. The paper evaluates different fusion techniques to combine information from RGB and Depth. These approaches are evaluated on a cleaned version of the NYUv2 dataset. The paper shows that on this cleaned NYU depth dataset, using a hybrid approach of combining information in the input space, and combining in the feature space works better than just using RGB. On the original dataset (that was noisier, and had descriptions for diverse objects from diverse viewpoints), depth fusion fails to produce better captions than RGB (as evaluated by automatic metrics).

**Audience:**

No

**Claims And Evidence:**

No

**Requested Changes:**

- Provide CIDER and SPICE metrics for evaluation.
- Test approaches using predicted depth.
- Evaluate approaches that leverage pre-training data (GIT, UNIFIED-IO, etc).
- Describe the cleaning process in more details.

**Strengths And Weaknesses:**

Strengths:

- The authors test an interesting hypothesis in this paper. Existing approaches in image captioning use only RGB input, and using Depth input to capture spatial and geometric knowledge about the scene is quite interesting.
- The authors exhaustively cover various techniques to fuse RGB and Depth images starting from early fusion in the input space, to late fusion in the feature space.

Weaknesses:

1. Choosing NYUv2 dataset feels like an odd choice to evaluate  — the dataset only covers indoor images, and it’s evaluated on a very small set of categories (21 object categories). Given the current state of AI, which contains much bigger benchmarks like COCO, nocaps, that is more diverse and span a much richer vocabulary of visual concepts, it feels restrictive to limit the evaluation to NYU depth.
2. The relabeling effort also seemed catered to make the hypothesis true — it concentrates on a few sets of categories rather than capturing the diversity of all objects present in the images. Second, relabeling focuses on describing the object’s appearance and 2D and 3D spatial relationships between objects. Other image captioning benchmarks collected captions with much looser guidelines, likely capturing more natural descriptions.
    - The evaluation benchmark should focus on capturing all the objects present in the dataset, irrespective of how many times they occur in the dataset.
    - Furthermore, no details on how the cleaned version of the dataset was created is presented in the paper? Was it collected using Amazon Mechanical Turk? Was the data cleaned in-house by the authors? How many captions were generated for each image? How many total images were used for training, and for validation and testing?
3. The metrics used for evaluation do not contain CIDEr and SPICE which are consistently used as standard metrics for evaluating image captioning methods.
4. The baselines used in the paper are weak and doesn’t leverage all the recent advances in vision and language. Do these methods benefit from large-scale pre-training (e.g, evaluating recent state of the art V+L methods), especially those that can take both RGB and Depth as input. What happens when such methods are fine-tuned on this data (RGB, Depth, Captions).
5. I would have also liked to see some experiments with using predicted depth instead of GT depth. For instance, using a recent depth model like MiDAS instead of GT depth.
    1. More importantly, another way to test the hypothesis of whether depth helps image captioning could be to use predicted depth along with RGB and evaluate on existing image captioning benchmarks.
    2. The hypothesis tested in this paper is interesting. But getting RGB data is *much* cheaper than getting depth data. Hence, relying on predicted depth as an auxiliary signal is quite interesting and should be evaluated to test this hypothesis.

---

> ### Author Response · Authors · 2023-03-16
> **Response to Reviewer thm9**
>
> Thank you for your comprehensive review and valuable feedback! We address your comments one by one as follows:
>
> ### 1. Provide CIDER and SPICE metrics for evaluation
>
> Thank you for this suggestion. We have incorporated your suggestions and added CIDEr results in Tables 2, 8, and 9 of our revised manuscript. Our model has achieved the highest improvement in CIDEr compared to other metrics, with an impressive increase of 8.9 points for the NYU-v2 dataset with GT depth, 7.9 points for the NYU-v2 dataset with predicted depth, and 3.01 points for the Stanford image paragraph captioning dataset with predicted depth.
>
> ---
>
> ### 2. Test approaches using predicted depth
>
> Thanks for this comment! Kindly refer to the detailed response for this point in the [common reply (part 1 - point 1)](https://openreview.net/forum?id=PtrK8Aoe2M&noteId=j354ozGi7U) under the ‘**Test the proposed approaches using predicted depth only**' title.
>
> ---
>
> ### 3. Evaluate approaches that leverage pre-training data (GIT, UNIFIED-IO, etc)
>
> That is an interesting perspective, so we are thankful for the comment. We do agree that evaluating recent state-of-the-art V+L would be very interesting. However,  our main goal was not to achieve state-of-the-art final performance. Rather, we aimed to investigate what would be the best approach to integrate the depth signal into the image captioning process. We have added this clarification in the section on baseline models, and we plan to evaluate approaches that leverage pre-training data in the future.
>
> ---
>
> ### 4. Describe the cleaning process in more details
>
> Thanks for this comment! Kindly refer to the detailed response for this point in the [common reply (part 1 - point 2)](https://openreview.net/forum?id=PtrK8Aoe2M&noteId=tql28zYAMs) under the ‘**NYU-v2 dataset cleaning**’ title.
>
> Please let us know if you have any further questions or suggestions.

---

### Review · Reviewer_nNXH · 2023-03-06

**Summary Of Contributions:**

The paper investigates a family of CNN/Transformer-based encoder-decoder models for multi-sentence caption generation from RGB-D images. The proposed network consists of four types of modules, including an image-depth fusion module, feature extractor backbone network, Transformer-based encoder and BERT decoder. To determine a proper model architecture, the paper introduces three types of image-depth fusion strategies with different variants in each type, different feature extractor backbones for image and depth, and two kinds of Transformer-based encoders. To evaluate the models, this work also develops a cleaned version of NYU-v2 dataset focusing on 21 object classes. The experimental evaluation compares the proposed method with several RGB baselines. It also performs an extensive study on different choices for each model component.

**Audience:**

Yes

**Broader Impact Concerns:**

It would be useful to show some statistics of the proposed dataset, and add a discussion on the potential bias in the dataset as well as the limitation of the work.

**Claims And Evidence:**

No

**Requested Changes:**

Please address the questions in the weaknesses above.

**Strengths And Weaknesses:**

Strengths:
- The problem of multi-sentence caption generation from RGB-D images is an interesting and novel problem.
- The paper presents a cleaned version of NYU-v2 dataset, which can be beneficial for research in image caption generation.
- The experiments provide an extensive study on different combinations of model components under the proposed model architecture.
- The paper is mostly well-written and easy to follow.

Weaknesses:
- The proposed problem setting seems a bit narrow/restrictive (multi-sentence+RGBD) and is not well motivated in the paper. While the RGBD input seems interesting and relevant in practice, it is unclear why it has to be multiple sentences. The specific property and challenges of multi-sentence captioning are not discussed in detail.
- The NYU-v2 dataset is relatively small and limited to indoor scenes. As such, its value to research in image captioning seems limited. What if the method is evaluated on outdoor scenes (e.g., MSCOCO) with predicted depth from pretrained image-to-depth models?
- The technical contribution of this work is limited. Most of the model components are taken from existing works and the main results are from the study on different combinations of those modules. However, the research methodology adopted in this work is a bit simple, which basically performs local search in the model architecture space. Such an approach lacks principle and may lead to sub-optimal conclusions. A better way seems to be adopting a search strategy as in Network Architecture Search (NAS).
- More importantly, it is unclear how the model selection is performed. Is it based on a validation set or on the test set directly? For the analysis in Sec. 6, it uses the test set to study different model configurations, which can lead to over-estimation of performance due to overfitting.
- The experimental comparison with the baselines is insufficient (Table 2). All the baselines only take RGB images as input, which is unfair. A more convincing comparison should use those models re-trained with RGB-D input.

---

> ### Author Response · Authors · 2023-03-16
> **Response to Reviewer nNXH**
>
> Thank you for your comprehensive review and valuable feedback! We address your comments one by one as follows:
>
> ### 1. Why it has to be multiple-sentences
>
> Thanks for pointing out this. Compared to single-sentence captioning, multi-sentence captioning is more challenging due to longer, more informative, and linguistically complex paragraphs. We believe that multi-sentence captioning is more suitable for leveraging depth information in RGB-D scenes, as it allows for a more detailed description of the 3D depth relations between objects. Single-sentence captioning, on the other hand, is more general in its description of the scene, and rarely involves describing precise spatial relationships between the objects, that is why we focus on multi-sentence captioning on this work
>
> ---
>
> ### 2. Test approaches using predicted depth
>
> Thanks for this comment! Kindly refer to the detailed response for this point in the [common reply (part 1 - point 1)](https://openreview.net/forum?id=PtrK8Aoe2M&noteId=j354ozGi7U) under the ‘**Test the proposed approaches using predicted depth only**' title.
>
> ---
>
> ### 3. Limited technical contribution
>
> Thanks for this comment! Kindly refer to the detailed response for this point in the [common reply (part 2- point 3)](https://openreview.net/forum?id=PtrK8Aoe2M&noteId=tql28zYAMs) under the ‘**Limited technical contribution**' title.
>
> Regarding the search strategy, we appreciate the suggestion of using a search strategy like Network Architecture Search (NAS). It is an interesting perspective. While we agree that using different search strategies, such as NAS, would be a worthwhile trial, our main goal in this work was to demonstrate the importance of depth for captioning. In future work, we would like to explore the benefits of using NAS, and we believe this could lead to even better performance.
>
> ---
>
> ### 4. How the model selection is performed
>
> Thanks for this comment! For the NYU-v2 dataset, we followed the official split employed by [1], where the images are divided into a training set and a testing set. The training set includes 795 scenes, while the testing set includes the remaining 654.
>
> For the Stanford image paragraph caption dataset, we followed the official split employed by [2], which divides the images into a training set, a validation set, and a testing set. The training set includes 14,575 images, the validation set has 2,487, and the testing set includes the remaining 2,489.
>
> ---
>
> ### 5. RGB-D baselines
>
> We would like to stress that our main goal in this work was to investigate whether we can exploit the depth signal for better image captioning and what would be the best approach to integrate the depth signal. That is why our experiments mainly focus on showing gains against the same RGB baseline. Thus, modifying and re-training other methods from literature wouldn’t be relevant for our goal here.
>
> Please let us know if you have any further questions or suggestions.
>
> ---
>
> [1] Lin, Dahua, Sanja Fidler, and Raquel Urtasun. "Holistic scene understanding for 3d object detection with rgbd cameras." Proceedings of the IEEE international conference on computer vision. 2013.\
> [2] Krause, Jonathan, et al. "A hierarchical approach for generating descriptive image paragraphs." Proceedings of the IEEE conference on computer vision and pattern recognition. 2017.

---

### Author Response · Authors · 2023-03-16
**Common reply (Part 1)**

We would like to thank all the reviewers for their constructive comments that have improved our manuscript. Please see the detailed responses for the common issues below, and let us know if you have any further questions or comments.

 ### 1. Test the proposed approaches using predicted depth only

To test the robustness of our proposed approach against using estimated / predicted depth-maps, we trained our proposed model on the Stanford image paragraph captioning dataset [1], which contains 19551 images, with predicted depth maps. Additionally, we trained our proposed model on the NYU-v2 dataset using predicted depth-maps instead of ground truth depth-maps. We use the DPT-Hybrid model to predict the depth maps.

Our experimental results, presented in Table 8 and Table 9 of our revised manuscript, demonstrate that our model can significantly outperform the RGB-only baseline in the captioning task, even with an estimated depth signal. This is particularly notable when using the feature-level fusion approach with the cross-attention method.

Table 8. Comparison of 3D captioning results obtained by our model and the baseline model on the Stanford image paragraph captioning dataset. **Bold** indicates the best results.

|                                   | Inputs     | Method                        | B-4  | R-1   | R-2   | R-L   | M     | C     |
|-----------------------------------|------------|-------------------------------|------|-------|-------|-------|-------|-------|
| Region-Hierarchical [1]           | RGB        | -                             | 8.69 | -     | -     | -     | 15.95 | 13.52 |
| Our 2D baseline(pre-trained BERT) | RGB        | -                             | 8.53 | 35.77 | 11.31 | 27.56 | 13.89 | 13.6  |
| Ours 3D (Pixel-level)             | RGB+Depth  | DMF                           | 7.71 | 36.67 | 12.32 | 27.46 | 13.14 | 15.47 |
| Ours 3D (Feature-level)           | RGB+Depth  | Late-fusion (Cross-attention) | **8.66** | **37.89** | **13.17** | **28.27** | 13.74 | **16.61** |
| Ours 3D (Hybrid)                  | RGB+MAE_CD | Late-fusion (Concat)          | 8.39 | 36.71 | 12.56 | 27.81 | 13.4  | 14.74 |

---
\
Table 9. Comparison of 3D captioning results obtained by our model and the baseline model on the NYU-v2 cleaned dataset with predicted depth instead of ground truth depth. **Bold** indicates the best results.

|                                   | Inputs     | Method                        | B-4       | R-1       | R-2       | R-L       | M         | C         |
|-----------------------------------|------------|-------------------------------|-----------|-----------|-----------|-----------|-----------|-----------|
| Our 2D baseline(pre-trained BERT) | RGB        | -                             | 10.99     | 44.5      | 19.56     | 41.04     | 30.25     | 17.54     |
| Ours 3D (Pixel-level)             | RGB+Depth  | DMF                           | 10.24     | 41.46     | 17.57     | 38.49     | 29.46     | 18.73     |
| Ours 3D (Feature-level)           | RGB+Depth  | Late-fusion (Cross-attention) | **12.54** | 45.03     | **19.83** | **41.72** | 32.27     | **25.53** |
| Ours 3D (Hybrid)                  | RGB+MAE_CD | Late-fusion (Concat)          | 10.79     | **45.16** | 18.98     | 41.24     | **32.95** | 23.03     |
---

These results are _**significant**_ because predicted depth information is much easier to obtain  and can still significantly enhance the quality of captions generated compared to using RGB data only. Using a predicted depth generated from a pre-trained depth estimation model, our framework can be widely applied to any RGB captioning dataset. In the future, we intend to explore the use of other types of signals to further improve the performance of the captioning model.

[1] Krause, Jonathan, et al. "A hierarchical approach for generating descriptive image paragraphs." Proceedings of the IEEE conference on computer vision and pattern recognition. 2017.

---

### Author Response · Authors · 2023-03-16
**Common reply (Part 2)**

### 2. NYU-v2 dataset cleaning

We have identified two issues in the original dataset. Firstly, it has inconsistent labelling. By "inconsistent labelling," we mean that while the dataset contains images of the same object, it is described in some images but not in others. Table 1 in the paper shows how many times each object class is mentioned in the text to all visual occurrences of that class, which proves this problem. The other problem was the non-uniform point of reference while describing the scene. Sometimes it is from the viewpoint of the viewer, and other times it is from the perspective of the described object.

These inconsistencies lead to confusion and inefficiency in the captioning data. Also, they impact the depth signal's capturing and prevent the model from learning from the depth information, as demonstrated by the results in Table 10.

We cleaned the dataset in-house, with each image having one caption, just like in the original version of the dataset. Figure 5 shows examples of the dataset before and after the cleaning step. We followed the partition employed by [2], where the images are divided into a training set and a testing set. The training set includes 795 scenes, while the testing set includes the remaining 654.

As shown in Figure 5, there were 4 cases while cleaning the dataset:
1. We removed descriptions of objects that were not part of the interest classes (e.g., fan, ironing board).
2. We added descriptions to objects that belong to the interest classes if they were missing (e.g., cabinets).
3. We changed the point of reference to unify it with the viewer.
4. No changes were made to the description if it already satisfied the conditions.

1380 images were modified overall after cleaning, leaving 69 images that remained unchanged. We added more details to clarify this point in the dataset cleaning subsection 4.1.

The cleaning process ensured that only the relevant objects were included in the dataset and that their descriptions were accurate and consistent. We believe this is crucial to any data, as the accuracy and consistency of the data can significantly impact the results.

Our cleaned version of the NYU-v2 dataset is now available at https://anonymous.4open.science/r/Cleaned-NYUv2-018A/

---
### 3. Limited technical contribution

We respectfully disagree with the reviewer and we believe that using a precise combination of a large number of techniques from the literature doesn’t make the contribution limited. Especially if the problem is not well explored and only a few select works in the literature study it. To the best of our knowledge, no other work in the literature has explored RGB-D image captioning using
- Cross attention to mix RGB and depth maps
- MultiMAE pre-trained representation for RGB and depth maps
- Systematic evaluation of where should signal fusion happen
- Demonstrating real gains with the depth signal, even if estimated with a pre-trained model

Also, looking in the literature, works like MaskRCNN [3], convNext [4], and GPT-3 [5] (just to name a few) made significant technical contributions while mainly employing and combining existing methods to get a new perspective on their respective problems .

Thank you again for your reviews - we hope our response clarifies your questions and concerns, but please let us know if you have any more!

---
[2] Lin, Dahua, Sanja Fidler, and Raquel Urtasun. "Holistic scene understanding for 3d object detection with rgbd cameras." Proceedings of the IEEE international conference on computer vision. 2013. \
[3] He, Kaiming, et al. "Mask r-cnn." Proceedings of the IEEE international conference on computer vision. 2017. \
[4] Liu, Zhuang, et al. "A convnet for the 2020s." Proceedings of the IEEE/CVF Conference on Computer Vision and Pattern Recognition. 2022. \
[5] Brown, Tom, et al. "Language models are few-shot learners." Advances in neural information processing systems 33 (2020): 1877-1901.

---

### Decision · Action_Editors · 2023-04-24

**Recommendation:** Reject

**Comment:**

Overall, the paper did not receive strong support after the response. The remaining issues were:

1. The question about model selection was not directly answered (Rev. nNXH).
2. The results on Stanford dataset are only on the level of the original paper from 2017 (13-15 CIDEr), whereas recent works have CIDEr more than doubled (30-36). Thus, it's unclear how applicable is the proposed method to recent captioning architectures. Incorporating depth information into these newer architectures will help show the suitability of the method (or on the opposite, its limitations).
3. The data collection process does not use best practices (Rev thm9):
  - The authors only collect one caption for each image instead of multiple captions (which helps the reported metrics to be more robust).
  - The data collection effort cleaned many categories which were inconsistently described in the dataset (See Figure 5, b in the main paper). Lamps and T-shirts are perfectly described in the image. However, since they might not have exhaustive annotations throughout the dataset, they were cleaned. This severely restricts the usefulness of the dataset. As the field is moving towards "open-vocabulary" detection and segmentation, I am a little disappointed that this new data only describes very few classes in the image, and actually "cleans" the original description to remove mentions of certain categories.
  - The data collection was done in-house, which can often lead to biases. If the people writing the paper, or testing specific hypotheses are the ones collecting the data, then it's problematic. The data will obviously start reflecting the hypotheses that it was being collected for.
4. Related to the above, there is a confounding factor in the experiments using NYUv2 (Table 10) since both the number of objects and the point of reference are changed. Thus, it is unclear if the improvement due to depth is a result of: a) limiting the scope to certain objects where depth is important; b) making the point of reference consistent so that depth can be better utilized.  Only fixing the point of reference of the original captions, and running the model could help better show the advantages/disadvantages of using depth.
5. It would have been more convincing if the method was evaluated on standard image captioning benchmarks like MS-COCO, nocaps, etc. Without which, the authors are testing a very narrow hypothesis -- depth information helps describe spatial relationships between objects in an image. (Rev thm9)

Due to the above issues, the paper requires a major revision and thus is not yet ready for publication in TMLR. The authors are encouraged to revise the paper accordingly.

**Audience:**

While the idea is interesting, the paper will be of limited appeal to TMLR audience due to limitations in the experiments. The paper's interest could be increased by expanding the experiments to more common captioning datasets and making the data collection of the NYUv2 dataset to follow best practices. Furthermore, incorporating depth into more recent SOTA methods could better show its applicability (or limitations).



**Claims And Evidence:**

The hypothesis that using depth information can improve captioning needs to be further tested. It would be helpful to see experiments on larger (common) datasets using predicted depth, e.g., MSCOCO.  Furthermore, the re-annotation of NYUv2 is not well designed and quite limited. Thus, further experiments are needed to assess the applicability of incorporating depth to various captioning domains (indoors vs. outdoors; limited number of objects vs. not, etc; sentence vs. paragraph). Adding these experiments will more thoroughly test the hypothesis and show when depth could be useful, and when it could not. The claims could then be adjusted accordingly.